# Genomic and bioacoustic variation in a midwife toad hybrid zone: A role for reinforcement?

**Johanna Ambu, Christophe Dufresnes**[ID][¤]*

Laboratory for Amphibian Systematics and Evolution, College of Biology & the Environment, Nanjing Forestry University, Nanjing, China

¤ Current address: Institut de Systématique, Evolution, Biodiversité (ISYEB), Muséum National d'Histoire Naturelle, CNRS, Sorbonne Université, EPHE-PSL, Université des Antilles, Paris, France
* Christophe.Dufresnes@hotmail.fr

**Data Availability Statement:** 16S sequences are available on GenBank (accessions OR725791–OR725973 and OR916149–OR916186); raw ddRAD sequence reads are archived on NCBI SRA under BioProject PRJNA949685; analyzed datasets

## Abstract

Hybrid zones, i.e., geographic areas where diverging lineages meet, hybridize and eventually mix their genomes, offer opportunities to understand the mechanisms behind reproductive isolation and speciation. Hybrid zones are particularly well suited to study reinforcement, i.e., the process by which selection against hybridization increases reproductive barriers, which, in anuran amphibians, is typically expressed by increased divergence in advertisement calls–the main cue to assortative mating–in parapatric ranges. Using mitochondrial barcoding (16S sequences), population genomics (thousands of SNPs) and bioacoustic analyses (four call parameters), we examine the hybrid zone between two incipient species of midwife toads (*Alytes obstetricans* and *A. almogavarii*) in southern France, with the purposes of locating their transition, measuring genetic introgression, and documenting potential signatures of reinforcement. We map range boundaries in the Eastern Pyrenees and the southwestern foothills of the Massif Central, namely along the Ariège valley and the Montagne Noire area. Similarly to another transition between these species in Spain, we found the hybrid zone to be narrow, involving geographically restricted gene flow (~20 km wide allele frequency clines) and barrier loci (i.e., loci resisting introgression), both suggestive of partial post-zygotic isolation (hybrid incompatibilities). The calls of the species overlap less inside than outside the hybrid zone, due to a reduction of their standing variation rather than a shift towards distinctive variants. While neutral causes cannot be excluded, this pattern follows the general expectations of reinforcement, yet without reproductive character displacement. Our study highlights the potential of amphibian hybrid zones to assess the genetic and behavioral drivers of reproductive isolation *in statu nascendi* and under various evolutionary contexts.

## Introduction

Reinforcement, i.e., the process by which reproductive barriers between incipient species are strengthened by natural selection, is one of the most discussed mechanisms of speciation [1–

(16S alignment, SNP data, bioacoustic data) are available on Zenodo (doi: 10.5281/zenodo.10037218).

**Funding:** This study was funded by the Taxon-Omics priority program (SPP1991) of the Deutsche Forschungsgemeinschaft (grant N° VE247/19-1 to CD) and by the National Natural Science Foundation of China (RFIS grant N° 3211101356 to CD). The funders had no role in study design, data collection and analysis, decision to publish, or preparation of the manuscript.

**Competing interests:** The authors have declared that no competing interests exist.

6]. Reinforcement typically acts in secondary contact zones and involves an increase of phenotypic differentiation between species that enhances their ability to avoid and/or discriminate each other via pre-mating barriers, and thus escape the fitness costs associated with maladaptive hybridization. Reinforcement has found support among ubiquitous taxa, including vertebrates (e.g., mammals [7]; birds [8–10]; fishes [11–13]; amphibians [14–16]), invertebrates (e.g., insects [17–28]), and plants [29–33]. Although reinforcement has received ample empirical and theoretical consideration [5, 34–37], its incidence and mode of operation remain elusive in many organisms.

Reinforcement in animals has been preferentially measured by comparing inter-specific behavioral differences in areas of sympatry *vs* areas of allopatry, in the form of reproductive character displacement, i.e., a disruptive shift in the traits involved in mate recognition (Fig 1, left). The signal may be complex, involving sex-specific and species-specific asymmetries, depending on a variety of factors such as the reproductive investment of females compared to males, the differential fitness of reciprocal cross directions, or the relative population sizes of the interacting species [5, 27, 34]. Alternatively, species differences may be reached without displacement (Fig 1, right). For polymorphic species, non-overlapping reproductive traits can be achieved by reducing the existing variation rather than evolving towards new phenotypes [34, 35].

Amphibians have long been considered a promising model to study reinforcement [3, 38] and some seminal work has demonstrated reproductive character displacements on coloration (e.g., poison frogs [39]; *Lissotriton* newts [40]) and advertisement calls (e.g., North American Chorus frogs [41, 42]; Australian tree frogs [43]). In anurans (frogs and toads), many species use vocalizations for mate recognition [44, 45], sometimes with high intra- and inter-specific variation [46]. Most of this variation is under genetic control and thus heritable [47, 48]. In parallel, many anuran lineages meet in secondary contact zones from where to characterize the onset of genetic and behavioral reproductive barriers in concert [49].

Here we examine genetic and bioacoustic variation in a hybrid zone between two incipient species of midwife toads, *Alytes obstetricans* and *A. almogavarii*. In midwife toads, mate

**Non-overlapping by character displacement**

**Non-overlapping by variance reduction (without character displacement)**

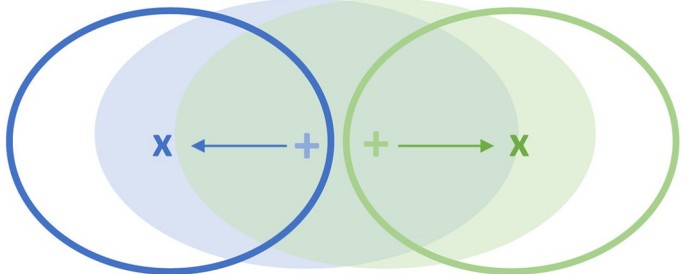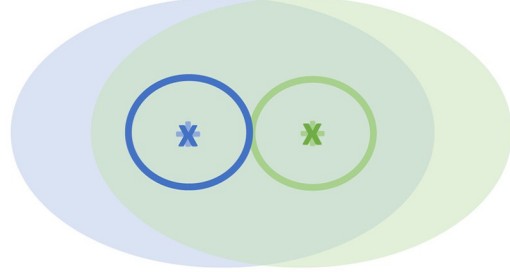

allopatric 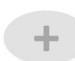 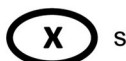 sympatric/parapatric

**Fig 1. Two hypotheses for the divergence of reproductive traits expected under reinforcement.** While allopatric populations of two species (plain ellipses and +) may retain widely overlapping variation for reproductive traits, reinforcement predicts a reduction of this overlap in the hybrid zone (empty ellipses and ×) as a selective response against maladaptive hybridization. This can be achieved by reproductive character displacement, which increases mean interspecific differences (left), or by reducing intraspecific variation without shifting the average phenotypes (right).

recognition relies on loud high-pitched vocalizations, and males invest as much reproductive effort as females due to a remarkable behavior of parental care that consists of nursing the eggs until they hatch [50]. The species under focus belong to a species complex that diversified in Western Europe within the last 4 My, with six phylogeographic lineages that retain high phenotypic (notably bioacoustic) polymorphism and meet in multiple contact zones [51, 52]. The previous analysis of one of these contact zones, between the subspecies *A. a. almogavarii* and *A. o. pertinax* in Catalonia, revealed geographically restricted admixture (with less than 20 km between the parental populations), which is consistent with introgressive hybridization mediated by intrinsic barriers to gene flow [53]. The partial reproductive isolation, similar lifestyle and rich bioacoustic repertoire thus offer a favorable context to document the evolution of reinforcement *in statu nascendi*.

In this study, we focus on a second hybrid zone in southern France, where the nominal subspecies *A. a. almogavarii* and *A. o. obstetricans* are expected to meet [52, 54]. Combining mitochondrial DNA barcoding, double digest Restriction Associated DNA sequencing (ddRAD-seq) and bioacoustic analyses, we locate the hybrid zone and test (1) whether admixture is constrained as in Catalonia, which would suggest that post-zygotic isolation between the two species also holds in this different phylogeographic setup; (2) whether the spectrum of mating calls overlaps less in the hybrid zone compared to the allopatric ranges, as expected under reinforcement; and (3) whether interspecies differences in the hybrid zone stem from character displacement.

## Materials and methods

### Genetic sampling

Tissue samples were collected for 227 individuals from 76 localities (S1 Table), either from live animals using buccal swabs (adults; [55]) or tail tips (tadpoles), or from vouchers curated in the scientific collections of BEV (CNRS–EPHE collection of the Biogeography and Ecology of the Vertebrates team in CEFE, Montpellier) and MNCN (Museo Nacional de Ciencias Naturales, Madrid). Sampling was approved by animal research ethics committees under collecting permits, namely the Direction Régionale de l'Environnement, de l'Aménagement et du Logement Occitanie (DREAL Occitanie, N˚DREAL-OCC-2023-s-05), the Reptielen Amfibieën Vissen Onderzoek Nederland (RAVON N˚FF/75A/2016/015) and the Direction Générale de l'Environnement, Division Biodiversité et Paysage (DGE-BIODIV, N˚2271). Tissue collection was non-destructive; live animals were captured, handled and released on site in less than a minute. Genomic DNA was isolated using an ad hoc salt protocol [56].

### mtDNA barcoding

We mapped the mtDNA lineages of *A. o. obstetricans* and *A. a. almogavarii* based on 619 individuals from 223 localities, combining published barcoding information [53, 54, 57, 58] with new sequences of the 16S rRNA gene for 221 of our samples (S1 and S2 Tables). To this end, a ~550 bp portion of 16S was amplified by PCR with primers 16SA-L and 16SB-H [59]. The reaction volumes (12.5μl) involved 8.05μl of water, 2.5μl of PCR buffer, 0.25μl of dNTPs (10μM), 0.3μl of each primer (10μM), 0.1μl of Taq polymerase and 1μl of DNA template. The thermocycling program consisted of an initial denaturation of 1'30" at 94˚C, 35 cycles of 45" at 94˚C, 45" at 53˚C, and 1' at 72˚C, followed by a final elongation of 1' at 72˚C. Amplicons were Sanger-sequenced in one direction (16SA-L). Raw sequences were quality-checked in MEGA X [60], before manual alignment and trimming (to 530 bp) in Seaview [61]. To identify the lineages, we reconstructed a maximum-likelihood (ML) tree with PhyML 3.0 [62].

## ddRAD-sequencing

We analyzed ddRAD-seq data for 89 samples from 41 localities (S1 Table). Samples were processed in genomic libraries prepared with a custom protocol adapted from [63] (https://dx.doi.org/10.17504/protocols.io.kxygx3nzwg8j/v1) and sequenced paired-end on an Illumina Next-Seq 550. Raw reads were demultiplexed with STACKS 2.59 [64] and trimmed to 2×65bp. The denovo_map.pl pipeline was applied for RAD loci construction, assembly, and cataloging with default settings. SNP calling was performed with the module *population* of STACKS to obtain the following datasets.

First, to infer the genome-average ancestry of samples, we produced a SNP matrix (*–structure*) by calling the loci shared by all 41 localities (*–p* 41) and by at least half of the samples of each locality (*–r* 0.5), while randomly keeping a single SNP per locus to avoid physically linked loci (*–write-random-snp*). The dataset contained 1,642 SNPs genotyped in the 89 samples (41 localities).

Second, to measure gene flow across the hybrid zone, we produced an allele frequency dataset along a 170 km west-east geographic transect (19 localities represented by 57 samples, S1 Table) for loci bearing fixed species differences. To identify such loci, we first selected 20 *obstetricans* and 19 *almogavarii* reference samples, i.e., free of introgression and broadly representative of the species ranges (S1 Table; see Results). We exported an allele frequency dataset (*–hzar*) of 53,458 SNPs present in both reference sets (*–p* 2) and in half of the samples of each (*–r* 0.5). Of these, we flagged 10,962 SNPs with fixed different alleles between the two groups (allele frequency difference of 1). To obtain an allele frequency dataset (*–hzar*) of the flagged loci for the transect populations, we used a whitelist (*–W*) and called only the loci genotyped in all transect localities (*–p* 19) and in at least half of the samples of each locality (*–r* 0.5), retaining only one SNP per locus (*–write-random-snp*). The obtained dataset consisted of 4,092 presumably unlinked species-diagnostic SNPs genotyped at the 57 samples (19 transect localities).

## Hybrid zone analyses

We inferred the nuclear ancestry from the 1,642 SNPs with the clustering program STRUCTURE 2.3.4 [65], using $K$ = 2. Several runs of 200,000 iterations after 20,000 burnin were performed to ensure convergence.

We quantified changes in allele frequency across the hybrid zone by fitting sigmoid clines on populations located along the transect using the $R$ package *hzar* [66]. We opted for models with two parameters (width $w$ and center $c$) in order to compare the clines obtained for: (1) the mitogroup frequencies; (2) the STRUCTURE population ancestry $Q$; (3) the allele frequencies of 4,092 species-diagnostic SNPs.

## Bioacoustic analyses

We recorded 406 advertisement calls from 71 individuals (2–9 calls per individual) spanning 25 localities (S1 Table). Recordings were made at night in June-July (the peak of breeding activity) with a PCM-A10 Sony recorder. Air temperature (T˚) was recorded and the body size (snout-vent length, SVL, measured with a 1 mm precision caliper) of individuals were measured whenever possible (n = 16 individuals from 7 localities), given their potential effects on temporal and spectral call parameters [67].

All recordings were processed in Raven Pro 1.6.1 (The Cornell Lab, Ithaca, NY, USA). Four parameters were measured on each note and averaged by individual: (1) the dominant frequency DF (Hz), as the frequency of the greatest energy in the note; (2) the note duration ND (s), as the time between the first and last pulses of the note, (3) the rising time RT (s), as the

time between the first pulse and the pulse with the highest frequency; (4) the pulse rate PR (s$^{-1}$), as the number of pulses per second. See S1 Fig for a graphical display.

SVL had no effect on any bioacoustic variable (Pearson's product-moment correlations, $P = 0.35–0.94$) but T° significantly influenced ND ($P < 0.005$, $r = -0.64$). To alleviate this effect, we used the residuals of the linear regression T°~ND in downstream analyses. Individuals were assigned four groups according to their species and geographic origin: allopatric *obstetricans* ($n = 27$), allopatric *almogavarii* ($n = 22$); parapatric *obstetricans* ($n = 9$); parapatric *almogavarii* ($n = 13$). Variation among groups was assessed by a Multivariate Analysis of Variance (MANOVA) and a Principal Component Analysis (PCA), building convex hulls to visualize group variation and overlap.

We performed three ad hoc statistical tests to explore specific expectations on the patterns of variation between species and allopatric/parapatric individuals under reinforcement and its underlying signatures (Fig 1). First, we compared hull overlaps between species, which under reinforcement should be lower for parapatric than allopatric individuals. To this end, we computed hull overlap with the *Overlap* function of the R package *shipunov* [68]. This was done separately for the two pairs of PCs (PC1/PC2 and PC3/PC4), and averaged over PCs, weighing for their relative contributions. Second, we compared hulls' centroid distances between species, which under character displacement should be higher between the parapatric than the allopatric individuals (left part of Fig 1). To this end, we measured the centroid (mean coordinate) of each group for each PC, weighted them by their relative contribution, and computed their pairwise Euclidian distance with the *dist* function of the R package *stats*. Third, we compared call diversity between the two species, which is expected to be lower among parapatric than allopatric individuals if between-species overlap is minimized by a reduction of the call repertoire (right part of Fig 1). To this end, we calculated the standard deviation of each group for each PC and weighed them by their relative contribution.

For each test, we computed relevant statistics, namely (1) the difference Δ between the allopatric and parapatric *almogavarii-obstetricans* overlaps; (2) the difference Δ between the allopatric and parapatric *almogavarii-obstetricans* centroid distances; (3) the differences Δ between the standard deviations of the parapatric and allopatric populations of each species, as well as the cumulated differences over both species. For statistical significance, we designed a permutation test that reshuffles group assignments and compute the Δ test statistics 1,000 times, in order to obtain their null distributions. The tests were considered significant when less than 5% of these null distributions exceeded the observed Δ (in absolute values). Because the sample sizes differed between groups, we performed a second set of permutations in which the two largest groups were randomly down-sampled to 10 individuals. All statistical analyses were conducted in R.

## Results and discussion

Mitochondrial barcoding and nuclear clustering analyses successfully distinguish *A. obstetricans* and *A. almogavarii* and provide concordant patterns regarding their respective distributions in southern France (Fig 2A and 2B; S2 Fig). We located at least two transitions: one in the Montagne Noire area (between locs 32 and 33) and one near the Ariège river in the Pyrenean foothills (between locs 10 and 11). The latter, which we densely sampled along a geographic transect, features traces of genetic admixture (and thus introgressive hybridization) on both sides of the river (locs 6–15). The mitochondrial and nuclear genome average clines have steep average widths $w$ of 9.8 km (95% confidence interval: 6.4–17.1) and 20.6 km (12.5–41.4), with roughly matching centers $c$ (~5 km apart) (Fig 2C, S3 Table). Clines inferred for individual SNPs follow the genome average ($w = 21.2 \pm 9.5$ km; Fig 2C and 2D) with relatively

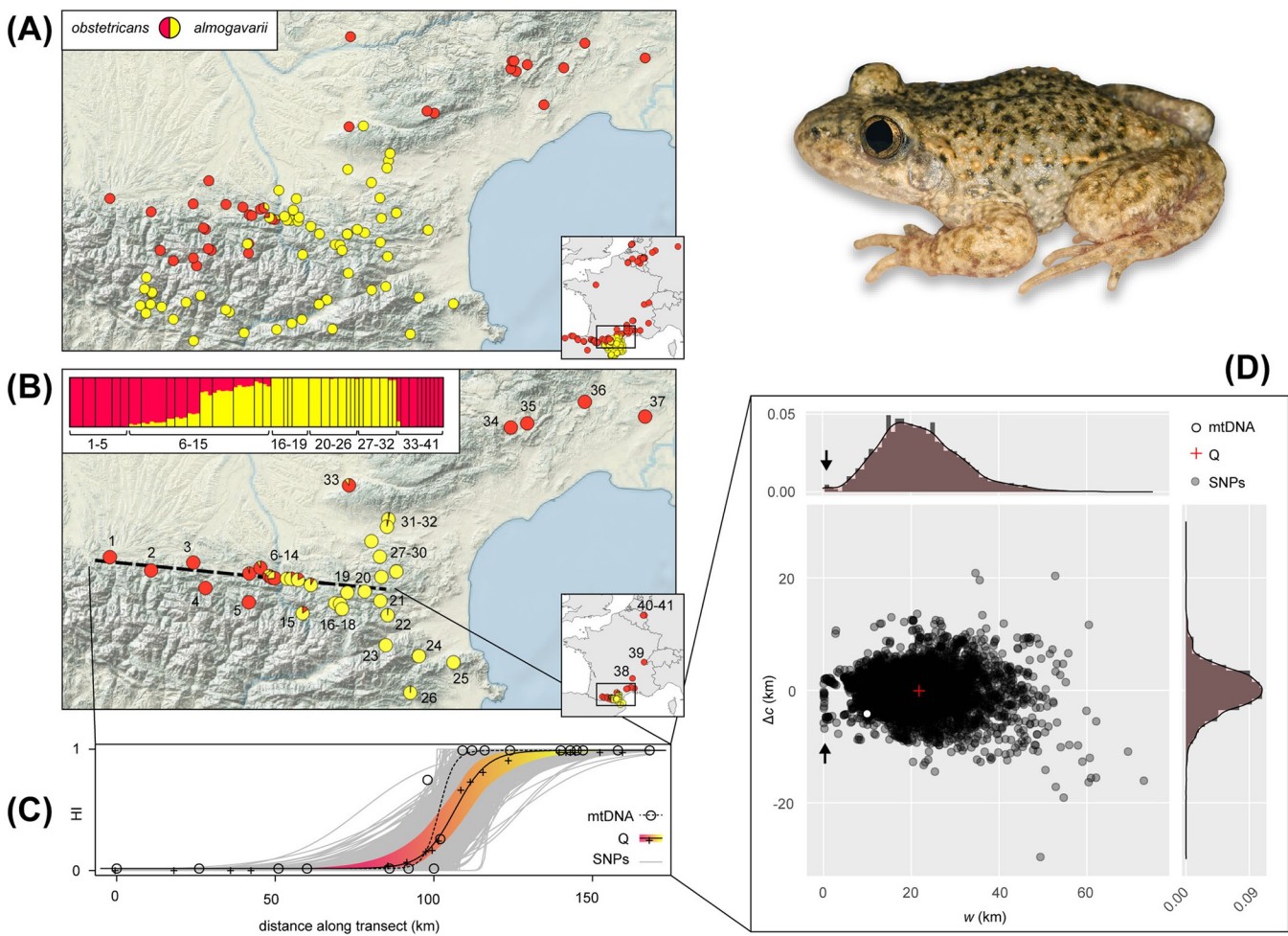

**Fig 2. Genetic structure of the *A. o. obstetricans* / *A. a. almogavarii* hybrid zone.** (A) Distributions of the mitochondrial lineages. (B) Individual (barplots) and population (piecharts on the map) nuclear ancestry estimated from 1,642 unlinked SNPs. (C) Sigmoid clines fitted on mitogroup frequencies (dashed cline), average ancestries (black plain cline, with the 95% confidence interval) and frequencies of 4,092 species-diagnostic SNPs (grey clines) along the geographic transect. (D) Variation in cline center (as the deviation $\Delta c$ from the average center) and cline width $w$. Arrows point to potential barrier loci (null width). Photo credit: J. Ambu.

homogeneous variation in both $w$ (~5–40km) and $c$ (+/- ~10km from the average $c$), except for a few SNPs where $w$ is essentially null (Fig 2D).

These results largely corroborate observations from the Catalonian hybrid zone, where the *A. obstetricans/almogavarii* transition is also characterized by steep clines (16 km on average) and a few loci entirely resist introgression [53]. These likely represent barrier loci, i.e., potentially linked to genes involved in Dobzhansky-Muller hybrid incompatibilities (as documented in other anuran hybrid zones [69]). Replicate hybrid zones between the same species pairs sometimes show contrasting outcomes, depending on intrinsic and extrinsic factors (e.g., [70, 71]). Instead, here *A. obstetricans* and *A. almogavarii* barely admix in both surveyed areas, regardless of the *A. obstetricans* lineages involved (subspecies *pertinax* in Catalonia vs. subspecies *obstetricans* in France), thus confirming their species status [72].

In parallel, we found significant differences in the mating calls of the two species (MANOVA, $F = 4.1$, dof = 66, $P = 0.005$). Most of these differences are driven by PC1 in the PCA (Fig 3), which reflects the pulse rate (PR) and the dominant frequency (DF)–both being

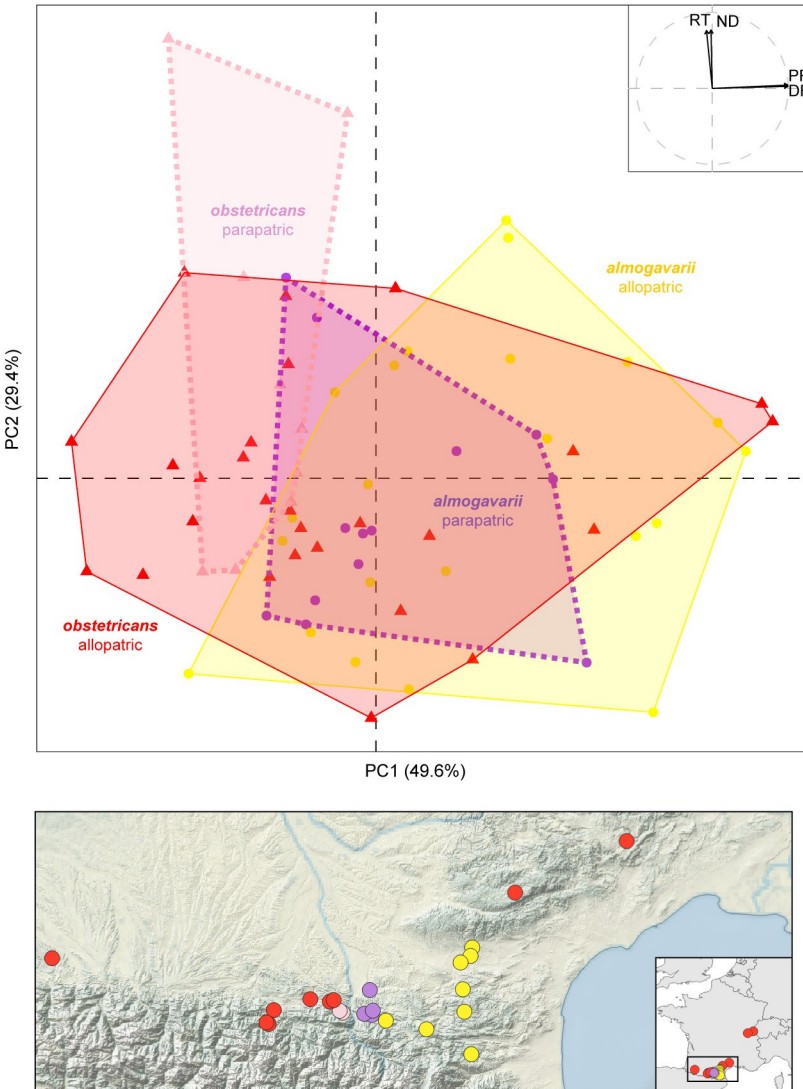

**Fig 3. Bioacoustic variation among 71 individuals in (parapatric) and out (allopatric) of the hybrid zone.** (A) PCA on four mating call parameters. (B) Geographic origin and group assignments of calls.

generally higher in *A. almogavarii* (S4 Table). The two species' calls significantly overlap less in the parapatric than the allopatric populations (Table 1, Fig 3). However, this reduced overlap is not due to a shift in the average call parameters: the interspecific call differences are not significantly greater between the parapatric and allopatric populations (Table 1). Instead, the parapatric individuals feature lesser call variation than the allopatric individuals, a trend that is marginally significant for *A. almogavarii*, and when combining both species together (Table 1). The same patterns are also obtained when the most represented groups are down-sampled in the permutation analyses (Table 1).

The genomic and bioacoustic variation observed thus comply with two of the general expectations of reinforcement: (1) that the hybrid zone is putatively maintained by partial reproductive isolation; and (2) that a reproductive cue (here, the advertisement calls) is more dissimilar between the two species in the populations from inside than outside of the hybrid zone. In addition, the dissimilarities reported here reflect a reduction of the species' standing

**Table 1. Comparison of the bioacoustic variation between allopatric and parapatric species across the four axes of the PCA (Fig 3).**

|  | allopatric | parapatric | Δ | *P* |
|---|---|---|---|---|
| Overlap between species | 0.82 | 0.33 | -0.49 | **0.022 (0.005)** |
| Centroid distance between species | 1.47 | 1.79 | +0.32 | 0.456 (0.269) |
| Standard deviation–*obstetricans* | 3.53 | 2.97 | -0.57 | 0.235 (0.271) |
| Standard deviation–*almogavarii* | 3.68 | 2.54 | -1.14 | **0.017 (0.047)** |
| Standard deviation–cumulated | 7.21 | 5.51 | -1.71 | **0.029 (0.048)** |

*P* indicates the proportion of the null distribution of Δ that exceeds the observed Δ (in absolute values); in bold when significant; brackets: *P* when down-sampling all groups to 10 individuals. For instance, only 2.2% (*P* = 0.022) of the 1,000 permutations yielded a greater allopatric-parapatric difference than the observed difference of 0.49, suggesting statistical significance.

variation (Fig 1, right) rather than a disruptive character displacement (Fig 1, left). Reinforcement is a form of local adaptation (to the presence of "wrong" mates). Therefore, assuming that mating call patterns in our midwife toad hybrid zone are driven by reinforcement, our observations are reminiscent of local adaptation literature, which accordingly assigns an important role to the fixation of existing variants rather than the evolution of new ones (e.g., [73]).

That being said, it is too preliminary to affirm that *A. obstetricans* and *A. almogavarii* are experiencing reinforcement, given that the documented bioacoustic variation may have also been influenced by other processes. Anuran calls are known to depend on various biotic and abiotic factors [45], related to e.g., meteorology [74], microhabitat [75], geography [46], predation pressure [76] and even anthropological disturbance [77]. Due to their geographic proximity, our parapatric populations inevitably share more similar environmental conditions than the more scattered allopatric populations, which could have led to the observed patterns of call variability. These patterns may also be a neutral consequence of introgressive hybridization: admixture in the parapatric populations homogenizes heritable phenotypes (thus preventing character displacement) while reducing individual fitness due to hybrid incompatibilities (thus decreasing effective population size and phenotypic variability). Future work should aim at quantifying assortative mating and the potential fitness costs of hybridization directly, as well as establishing causality links with the observed variation at mating calls, for instance, under experimental setups. Moreover, reinforcement is expected to leave peaks of genomic differentiation between sympatric and allopatric individuals that are shared across both species, notably around the genes involved in mate-recognition [78]. Such signature may be screened from our ddRAD-seq data, providing that the loci can be mapped to long-read assemblies such as a reference genome.

To date, reinforcement has been primarily examined in reproductively well-isolated species that form bimodal hybrid zones i.e., where hybridization and introgression is drastically restricted, and distinct hybrid/parental categories co-exist. Some of these cases have been criticized because character displacement between deeply-diverged species can be explained by alternative processes (e.g., ecological adaptations that evolved before the contact [5, 79]; runaway sexual selection [80]). Our study highlights that younger, lesser differentiated species that form unimodal hybrid zones, i.e., where large parts of the genomes remain permeable to gene flow, can in principle feature the genetic and phenotypic variation necessary to evolve reinforcement, even without obvious character displacement. The empirical focus on such systems should thus widen our understanding of the phenomenon across broader evolutionary contexts.

## Supporting information

**S1 Table. Details on the samples included in this study.** Column # indicates locality number as in Fig 1 (bold: transect samples; * reference samples to determine species-diagnostic RAD loci). Columns Y and X indicate latitude and longitude. Columns 16S and ddRAD-seq indicate individual accession numbers on GenBank and NCBI Sequence Read Archive (SRA). Column Call indicates individuals recorded for bioacoustic analyses, with reference to the online repository.
(DOCX)

**S2 Table. Numbers of individuals with the *A. o. obstetricans* and *A. a. almogavarii* mtDNA lineages in genetically barcoded localities, combining our study and the literature.**
(DOCX)

**S3 Table. Cline parameters *c* (center) and *w* (width) for the different datasets.** For mtDNA and Q, numbers correspond to the averages and 95% confidence intervals. For the species-diagnostic loci, numbers correspond to the median and standard deviation among the 4,092 SNPs.
(DOCX)

**S4 Table. Mean, standard variation (±) and range (in brackets) of the bioacoustic variables in the four groups.** DF: dominant frequency; ND: note duration; RT: rising time; PR: pulse rate.
(DOCX)

**S1 Fig.** Oscillogram (top) and spectrogram (bottom) of a note, showing the variables measured. PR is obtained by counting the number of P pulses per time unit (dividing by ND).
(DOCX)

**S2 Fig. ML phylogenetic analysis of the 16S sequences for DNA barcoding.**
(DOCX)

## Acknowledgments

We are grateful to O Buisson, P-A Crochet, C Delmas, M. Denoël, N Fuento, P Geniez, M. Gilbert, O Hadj-Bachir, P. Lemmers, I Martínez-Solano, T Suchan, A Trochet, M Vences and B Wielstra for discussion, lab support and sample collection/access.

## Author Contributions

**Conceptualization:** Johanna Ambu, Christophe Dufresnes.

**Formal analysis:** Johanna Ambu.

**Funding acquisition:** Christophe Dufresnes.

**Investigation:** Johanna Ambu, Christophe Dufresnes.

**Project administration:** Christophe Dufresnes.

**Supervision:** Christophe Dufresnes.

**Validation:** Christophe Dufresnes.

**Visualization:** Johanna Ambu.

**Writing – original draft:** Johanna Ambu.

Writing – review & editing: Christophe Dufresnes.

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
