## [Decision Letter · Decision Letter 0]

26 Aug 2024

PONE-D-24-26054Genomic and bioacoustic variation in a midwife toad hybrid zone: a role for reinforcement?PLOS ONE

Dear Dr. Dufresnes,

Thank you for submitting your manuscript to PLOS ONE. After careful consideration, we feel that it has merit but does not fully meet PLOS ONE’s publication criteria as it currently stands. Therefore, we invite you to submit a revised version of the manuscript that addresses the points raised during the review process.

Having considered the feedback from both reviewers, I am in agreement that the manuscript can be accepted after minor revisions.

We look forward to receiving your revised manuscript.

Kind regards,

Amaal Gh. Yasser, Ph.D.

Academic Editor

PLOS ONE

Journal Requirements:

2. Please clarify the issuing authority of the permits obtained, including spelling out any acronyms. Please also specified the fate of all animals used - whether released back in the environment or something else, and whether any animals died during the work.

"This study was funded by the Taxon-Omics priority program (SPP1991) of the Deutsche Forschungsgemeinschaft (grant N° VE247/19-1 to CD) and by the National Natural Science Foundation of China (RFIS grant N° 3211101356 to CD)."

5. We note that [Figures 2 and 3] in your submission contain [map/satellite] images which may be copyrighted. All PLOS content is published under the Creative Commons Attribution License (CC BY 4.0), which means that the manuscript, images, and Supporting Information files will be freely available online, and any third party is permitted to access, download, copy, distribute, and use these materials in any way, even commercially, with proper attribution. For these reasons, we cannot publish previously copyrighted maps or satellite images created using proprietary data, such as Google software (Google Maps, Street View, and Earth). For more information, see our copyright guidelines: http://journals.plos.org/plosone/s/licenses-and-copyright.

a. You may seek permission from the original copyright holder of Figures 2 and 3 to publish the content specifically under the CC BY 4.0 license.  

Additional Editor Comments:

Having considered the feedback from both reviewers, I am in agreement that the manuscript can be accepted after minor revisions.

Reviewers' comments:

Reviewer's Responses to Questions

**Comments to the Author**

1. Is the manuscript technically sound, and do the data support the conclusions?

Reviewer #1: Yes

Reviewer #2: Yes

2. Has the statistical analysis been performed appropriately and rigorously? 

Reviewer #1: Yes

Reviewer #2: Yes

3. Have the authors made all data underlying the findings in their manuscript fully available?

Reviewer #1: Yes

Reviewer #2: Yes

4. Is the manuscript presented in an intelligible fashion and written in standard English?

Reviewer #1: Yes

Reviewer #2: Yes

5. Review Comments to the Author

Reviewer #1: The study focuses on patterns of genomic and bioacoustic differentiation in a hybrid zone involving two species of midwife toads. The data are comprehensive and well suited to tests hypotheses about the role of reinforcement in the maintenance of species boundaries. The analyses are state of the art and the results are discussed appropriately, including potential shortcomings and future directions. I have made my suggestions, all minor, directly on the Word file (attached).

Reviewer #2: The paper is very interesting and well written. Hybrid zones provide good opportunities to understand mechanisms of reproductive isolation and speciation. The paper examines one of the important aspects - the reinforcement (by which reproductive barriers should be increase). Using mitochondrial barcoding, population genomics, and bioacoustic analysis, the authors studied the hybrid zone between two closely related amphibian species. It turned out that the breeding calls of the species overlap less inside than outside the hybrid zone. Perhaps, this is due to a reduction of their standing variation rather than a shift towards distinctive variants. This finding is very important for assessing the genetic and behavioral factors that contribute to reproductive isolation.

I did not find any inconsistencies. Therefore, I recommend it for publication in this journal.

6. PLOS authors have the option to publish the peer review history of their article (what does this mean?). If published, this will include your full peer review and any attached files.

Reviewer #1: No

Reviewer #2: No

---

## [Author Response · Author response to Decision Letter 0]

11 Oct 2024

Response to Referees

Copyright queries (sent 11 october 2024 by Richard Ibañez Dilla)

1. Thank you for confirming that the map images in Figures 2 and 3 were created with shapefiles from Natural Earth and are in the public domain. However, please note that we require additional information about the other image in Figure 2 (i.e. frog image). Before we proceed, please respond to the remaining copyright queries:

1) Please explain where the authors obtained the image of a frog in Figure 2 in your submission. If the authors created this image themselves, please confirm this in your response.

R: The image was created by us. We have now specified this in the legend of Figure 2 by adding the photo credit (J. Ambu, the first author).

2) Please state whether the image of a frog in Figure 2 has been previously published or copyrighted. Note: If the authors created this image themselves and this image has not been previously copyrighted, please disregard question (3) below.

R: We confirm that the image has not been previously published or copyrighted.

Journal Requirement queries:

R: We have accordingly revised our entire manuscript to match the journal’s style following the tutorial files above.

2. Please clarify the issuing authority of the permits obtained, including spelling out any acronyms. Please also specified the fate of all animals used - whether released back in the environment or something else, and whether any animals died during the work.

R: We have spelled out the issuing authorities of the permits and clarified that animals were released to their place of capture immediately after handling in the Material and Methods section.

"This study was funded by the Taxon-Omics priority program (SPP1991) of the Deutsche Forschungsgemeinschaft (grant N° VE247/19-1 to CD) and by the National Natural Science Foundation of China (RFIS grant N° 3211101356 to CD)."

R: The funders took no part in the study and we have added the above statement in our cover letter, as suggested.

R: We have now removed any other mention of the ethics statement apart from the Material and methods section, as requested.

5. We note that [Figures 2 and 3] in your submission contain [map/satellite] images which may be copyrighted. All PLOS content is published under the Creative Commons Attribution License (CC BY 4.0), which means that the manuscript, images, and Supporting Information files will be freely available online, and any third party is permitted to access, download, copy, distribute, and use these materials in any way, even commercially, with proper attribution. For these reasons, we cannot publish previously copyrighted maps or satellite images created using proprietary data, such as Google software (Google Maps, Street View, and Earth). For more information, see our copyright guidelines: http://journals.plos.org/plosone/s/licenses-and-copyright.

a. You may seek permission from the original copyright holder of Figures 2 and 3 to publish the content specifically under the CC BY 4.0 license. 

R: These maps were built by us based on background shapefile layers obtained from Natural Earth (public domain) so this is not an issue. In its terms of use, the Natural Earth website accordingly states “No permission is needed to use Natural Earth. Crediting the authors is unnecessary.”. We therefore did not change these figures nor their legend in our revision.

R: We have provided captions for SI files following the guidelines and double-checked that these match in-text citation, accordingly.

R: We have reviewed our reference list accordingly and did not add or remove any citation. 

Additional Editor Comments:

Having considered the feedback from both reviewers, I am in agreement that the manuscript can be accepted after minor revisions.

R: Thank you. We have conducted the requested minor revisions.

Review Comments to the Author

Reviewer #1: The study focuses on patterns of genomic and bioacoustic differentiation in a hybrid zone involving two species of midwife toads. The data are comprehensive and well suited to tests hypotheses about the role of reinforcement in the maintenance of species boundaries. The analyses are state of the art and the results are discussed appropriately, including potential shortcomings and future directions. I have made my suggestions, all minor, directly on the Word file (attached).

R: Thank you for your positive feedback and in-text comments. We have considered all of them in the revisions, with our specific answers provided in your file.

Reviewer #2: The paper is very interesting and well written. Hybrid zones provide good opportunities to understand mechanisms of reproductive isolation and speciation. The paper examines one of the important aspects - the reinforcement (by which reproductive barriers should be increase). Using mitochondrial barcoding, population genomics, and bioacoustic analysis, the authors studied the hybrid zone between two closely related amphibian species. It turned out that the breeding calls of the species overlap less inside than outside the hybrid zone. Perhaps, this is due to a reduction of their standing variation rather than a shift towards distinctive variants. This finding is very important for assessing the genetic and behavioral factors that contribute to reproductive isolation.

I did not find any inconsistencies. Therefore, I recommend it for publication in this journal.

R: Thank you for your positive feedback and recommendation of acceptance without revisions.

---

## [Editor Report · Decision Letter 1]

12 Nov 2024

Genomic and bioacoustic variation in a midwife toad hybrid zone: a role for reinforcement?

PONE-D-24-26054R1

Dear Dr. Christophe Dufresnes,

We’re pleased to inform you that your manuscript has been judged scientifically suitable for publication and will be formally accepted for publication once it meets all outstanding technical requirements.

Kind regards,

Amaal Gh. Yasser, Ph.D.

Academic Editor

PLOS ONE
---

## [Editor Report · Acceptance letter]

15 Nov 2024

PONE-D-24-26054R1 

PLOS ONE

Dear Dr. Dufresnes, 

I'm pleased to inform you that your manuscript has been deemed suitable for publication in PLOS ONE. Congratulations! Your manuscript is now being handed over to our production team.

Kind regards, 

on behalf of

Dr. Amaal Gh. Yasser 

Academic Editor

PLOS ONE